# Application of a Polysaccharide Purification Instrument—The Preparation and Characterization of Soybean Soluble Polysaccharide

**DOI:** 10.3390/polym17040480

**Published:** 2025-02-12

**Authors:** Xuhui Zhuang, Hongjuan Chen, Xiaohong Luo, Wei Han, Yongtan Yang

**Affiliations:** Academy of National Food and Strategic Reserves Administration, Beijing 100037, China; zxh@ags.ac.cn (X.Z.); chj@ags.ac.cn (H.C.); lxh@ags.ac.cn (X.L.)

**Keywords:** polysaccharide purification instrument, soybean soluble polysaccharides, preparation of polysaccharides, structural characterization

## Abstract

Polysaccharides in plants and microorganisms have important application value, and their purification and preparation is a prerequisite for in-depth research. However, there is currently a lack of dedicated separation and purification instruments for polysaccharide substances. In our previous work, a polysaccharide purification instrument (PSPI) was designed using post-column split-flow and post-column derivatization schemes and developed. In this study, the PSPI was applied to separation and preparation of the soybean soluble polysaccharides (SSPSs) and obtained the purified SSPS (SSPS-P). The total carbohydrate content in SSPS-P reached 97.2%, compared to 81.7% in SSPS, and the carbohydrate recovery rate was 86.5%. The composition and structure of SSPS-P have been assessed by HPLC, FT-IR, and NMR. SSPS-P was a polysaccharide with a molecular weight (M_w_) of 354 KDa, composed of D-glc, D-gal and L-ara with the molar ratio of 0.02:2.08:1.01. The structure of SSPS-P was mainly →4)-β-gal*p*-(1→unit. The α-L-ara*f* residues were also detected in the form of T-α-L-ara*f*-(1→2)-α-L-ara*f*-(1→, →3)-α-L-ara*f*-(1→ and →3,5)-α-ara*f*-(1→. PSPI could be applied for rapid and precise separation and preparation of polysaccharides.

## 1. Introduction

Polysaccharides have high conformational variability and possess many structures and biological information that proteins and nucleic acids cannot provide [1]. Therefore, they exhibit physicochemical properties such as water retention and emulsification [2], as well as physiological functions such as antioxidant, antibacterial, and antitumor activities [1]. Due to the complex structure, they need to be well characterized, thus high-throughput purification methods and instrumentation are necessary. Currently, the main methods for the separation and purification of polysaccharide compounds include ultrafiltration [3], fractional alcohol precipitation [3,4], ion exchange chromatography [3,4], and size exclusion chromatography (SEC) [3,4,5]. Among them, SEC is the most commonly used and efficient method. High-performance size exclusion chromatography (HPSEC), equipped with pressure-resistant gel and a high-pressure pump, can separate and prepare polysaccharides with different molecular weights [6] and can also analyze and calculate the molecular weight of polysaccharide samples [6]. However, due to the absence of chromophore in the molecule, a polysaccharide does not have characteristic UV-visible absorption bands, making it impossible to be quickly identified and separated from mixtures using a UV-Vis detector in preparative HPSEC. Currently, some universal detectors such as the differential refractive index detector (RID), evaporative light-scattering detector (ELSD), or multi-angle light scattering detector (MALSD) were broadly equipped in HPSEC and high-performance gel permeation chromatography (HPGPC); however, they could not distinguish the polysaccharides from other macromolecular substances. This can easily mislead the choices of target substances during polysaccharide purification and preparation.

A polysaccharide purification instrument (PSPI) has been designed using post-column split-flow and post-column derivatization schemes [7,8]-in our previous work, which enabled the precise separation and preparation of Dextran (Mw10000) in our previous test. However, this instrument has not been applied for the separation and preparation of a polysaccharide extracted from natural resources yet.

Soybean soluble polysaccharide (SSPS) is mainly extracted from soybean meal, which appears as a light yellow or white powder. The extracting solvent could be hot water, acidic water, or alkaline water. For example, the yield of SSPS was 16.24% under the extracting condition of pH 11.0, extraction temperature 120 °C, ratio of solid to liquid 1:20 (g:mL), and extraction time 2 h. However, the polysaccharide yield was up to 37.88% when it was extracted using acid water (pH 4.0) at the extraction temperature of 118 °C, ratio of material to liquid 1:30, and the extraction time of 2.5 h [9]. Many assisted methods such as ultrasonic and microwave are commonly used to further improve the yield of SSPS.

SSPS has been proven to decrease human plasma cholesterol levels as early as the year 1985 [10]. It also has the bio-functions such as regulating immunity and anti-tumor [11], antioxidant [12], promoting intestinal peristalsis [13,14], and reducing blood sugar [13], as well as the physical properties such as emulsification, foam stability, anti-caking, film forming, etc. [15,16]. Due to its properties of being easily soluble in water, acid resistant, and low viscosity, it is often used as a food additive or as film material [17,18]. Therefore, SSPS is a polysaccharide with great application value and development potential. The PSPI might provide an efficient pathway for the high-throughput and efficient preparation of SSPS. The chemical structure of SSPS was proven to be a hetero-polysaccharide which has a globular structure with long neutral side chains of α-(1→3) and α-(1→5)-arabinan and β-(1→4)-galactan [19]. The composition of arabinan and galactan were reported to be 20.7% and 49.8% of total sugars, respectively [19]. Nakamura et al. [20] found that the SSPS was composed of galacturonan backbone of homogalacturonan (α-1,4-galacturonan) and rhamnogalacturonan (repeating units being composed of α-1,2-rhamunose and α-1,4-galacturonic acid) branched by β-1,4-galactan and α-1,3- or α-1,5-arabinan chains. The composition and structure of SSPS might greatly vary with the extraction method applied.

In this article, a crude SSPS sample with a total sugar content of 81.7% was purified and prepared using the developed PSPI. The composition and structure of the purified SSPS (named SSPS-P) have been assessed by HPGPC, FT-IR, and NMR methods.

## 2. Materials and Methods

### 2.1. Chemicals and Materials

The crude soybean soluble polysaccharide sample (with a total sugar content of 81.7% tested using the phenol-sulfuric acid method) was provided by Northeast Agricultural University in Harbin, Heilongjiang province, China. The reagents used for the derivatization reaction were concentrated sulfuric acid and 6% Phenol solution. The concentrated sulfuric acid (H_2_SO_4_, Sinopharm Chemical Reagent Shanghai Co., Ltd., Shanghai, China) was of superior purity. A 6% Phenol solution (Ph) was prepared by dissolving every 6 g of phenol (AR grade, Sinopharm Chemical Reagent Shanghai Co., Ltd., Shanghai, China) in 100 mL water and then storing the solution away from light. Sodium nitrate (NaNO_3_, AR grade) was purchased from Sinopharm Chemical Reagent Shanghai Co., Ltd., Shanghai, China. Trifluoroacetic acid (TFA, HPLC grade), D-glucose (D-glc), D-galactose (D-gal), and L-arabinose (L-ara), as well as the dextran standards with the weight of molecular weight (M_w_) of 4320 g/mol, 12,600 g/mol, 60,600 g/mol, 110,000 g/mol, and 289,000 g/mol, were purchased from Macklin (Shanghai Macklin Biochemical Technology Co., Ltd., Shanghai, China). Water used for chromatography experiments and aqueous solution preparation was self-made laboratory-grade water (room temperature ρ = 18.25 MΩ × cm).

### 2.2. Working Principle and Setup of PSPI

The working principle of PSPI and the developed instrument is shown in Figure 1. Based on a traditional HPSEC, a splitter was added at the rear end of the chromatographic column, which allowed the effluent from the column to be divided into two flow paths: the Derivation Flow Path (DFPath) and the Collection Flow Path (CFPath). In the DFPath, the polysaccharide sample reacted with the derivatization reagent to generate derivatives which have the characteristic absorption peak when going through the UV-Vis detector. The CFPath could be detected by RID, and the target peak would be determined by comparing it with the DFPath. The target would be collected in the flow fraction collector, thus achieving precise separation and preparation of polysaccharides. The flow path of the instrument is as follows: The mobile phase (water) was pumped by a high-pressure pump (1) to a six-port valve (2), and the polysaccharide sample was injected into a quantitative loop (4) by a sample injection needle (3). It entered the chromatographic column (5) in the mobile phase, and the eluted components were divided into DFPath and CFPath by a splitter (6). The sample in the DFPath reacted with the derivatization reagent (concentrated sulfuric acid and 6% Phenol solution were pumped into the pipeline by pumps 7 and 8, respectively) in the derivatization device (9) to produce the reaction products which have characteristic absorption and can be analyzed by the UV-Vis detector (12), which were then discharged to waste liquid (16) after analysis. The CFPath passed through a pressure regulator (10) and was determined by RID (11) to identify the target, which was then collected and prepared by the flow fraction collector (14). The detection signals of RID and UV-Vis were transmitted to the control system (15) through a digital-to-analog converter (13) for synchronous monitoring.

In Figure 1b presented the schematic diagram of PSPI set up in this work based on a semi-preparative high-performance liquid chromatography (Agilent 218, Agilent Technologies, Inc., Santa Clara, CA, USA). In addition, Agilent 1260 Infinity II Refractive Index Detector (Agilent Technologies, Inc., Santa Clara, CA, USA) was performed in CFPath. Waters 515 high-pressure pump (Waters Corporation, Milford, MA, USA) was used to pump the mobile phase into the system. The derivative reagent pumps and the UV-Vis detector were purchased from Shanghai Sanotac, Inc. (Shanghai, China). The derivatization device and the pressure regulator were provided by Huapai Instrument (Qingdao) Co., Ltd. (Qingdao, China). Agilent 35900E (Agilent Technologies, Inc., Santa Clara, CA, USA) digital-to-analog converter was used to convert the digital signals of RID and UV-Vis detectors into analog signals then sent to the control computer. Agilent “Control Panel” workstation was used to monitor the system.

Dextran10000 (Shanghai Macklin Biochemical Technology Co., Ltd., Shanghai, China) was used for the modification and testing of the equipment. The retention time of Dextran10000 was adopted to calibrate the peak time of DFPath and CFPath using a TSKgel G3000PWxl gel exclusion column (7 μm, 21.5 × 300 mm, Tosoh Corporation, Yamaguchi, Japan). The lengths of the pipelines in the two flow paths were optimized to ensure that the retention time of Dextran10000 in both flow paths remains consistent. The instrument was verified by Shanghai WEIPU Testing Technology Group Co., Ltd. (Shanghai, China). as a third-party verification (Verification Number: CBJ-WP-BG-001-2023).

### 2.3. Methods for Purification and Preparation

A total of 10 mg/mL crude SSPS aqueous solution was prepared. We took 1 mL and then injected it into the PSPI equipped with a TSKgel G3000PWxl gel exclusion column (7 μm, 21.5 × 300 mm, Tosoh Corporation, Yamaguchi, Japan). The mobile phase was water with a flow rate of 1.2 mL/min. In the DFPath, the temperature for derivatization reaction was 80 °C, and the detection wavelength was 490 nm at room temperature. The target peak was identified by the retention time of the chromatographic peaks of DFPath and CFPath in Agilent Control Panel Chemstation (Agilent Technologies, Inc., Santa Clara, CA, USA). The target considered as SSPS-P was collected by a flow fraction collector equipped in the Agilent 218 semi-preparative high-performance liquid chromatography. SSPS-P was then freeze-dried and characterized. The pureness and the rate of recovery of total carbohydrate were determined using the phenol-sulfuric acid method.

### 2.4. Characterization of SSPS-P

#### 2.4.1. M_w_ Determination

A 1.0 mg/mL SSPS-P sample aqueous solution was prepared. The M_w_s of SSPS and SSPS-P were analyzed using a high performance liquid chromatography (HPLC, Waters 2695, Waters Corporation, Milford, MA, USA) equipped with RID (Waters 2414, Waters Corporation, Milford, MA, USA). Waters Ultrahydrogel^TM^ Linear column (7.8 × 300 mm, Waters Corporation, Milford, MA, USA) and Waters Ultrahydrogel^TM^ 250 column (7.8 × 300 mm, Waters Corporation, Milford, MA, USA) were used in series. Sample solutions (100 μL) were injected and run with 0.1 mol/L NaNO_3_ aqueous solution at 0.6 mL/min as the mobile phase. The column oven temperature was 40 °C. The standard curve was established by using dextran standards with M_w_s of of 4320 g/mol, 12,600 g/mol, 60,600 g/mol, 110,000 g/mol, and 289,000 g/mol. The molecular weight of each composition was calculated by contrast with the retention time of polysaccharides reference standard.

#### 2.4.2. Monosaccharide Composition Analysis

A total of 100 mg SSPS-P was hydrolyzed with 4 mL of 1 mol/L TFA for 90 min at 105 °C and then was dried using a rotary evaporator. The hydrolysate was solved with 10 mL water, and then 50 μL was taken to inject into the HPLC (Waters 2695) equipped with RID (Waters 2414). Aminex ^®^ HPX-87C (300 mm × 7.8 mm, Bio-Rad Laboratories, Inc., Hercules, CA, USA) sugar analysis chromatography column was used with the column temperature of 85 °C, which was regulated by an external column temperature box (temperature fluctuation ≤ ±0.5 °C). The mobile phase was water with a flow rate of 0.3 mL/min. Internal temperature of RID was set to be 50 °C.

#### 2.4.3. Fourier Transform Infrared Spectroscopy (FT-IR) Analysis

The functional groups of SSPS-P were analyzed by FT-IR [21]. SSPS-P was ground with KBr powder in a ratio of 1:100 and then pressed into a 1 mm pellet for FT-IR measurement. FT-IR spectra was obtained within a range of 4000–500 cm^−1^ using a Spectrum FT-IR spectrometer (Thermo fisher Nicolet iN10-iZ10, Thermo fisher, Waltham, MA, USA).

#### 2.4.4. Nuclear Magnetic Resonance (NMR) Analysis

SSPS-P saturated solution was prepared with 0.5 mL of Deuterium oxide (D_2_O, Cambridge Isotope Laboratories, Inc, Tewksbury, MA, USA). The supernatant was centrifuged and taken into an NMR tube (Norell, purchased from Tenglong Weibo Tech. Co., Ltd., Qingdao, China). ^1^H-NMR, ^13^C-NMR, 135° distortionless enhancement by polarization transfer (135 DEPT), total correlation spectroscopy (TOCSY), heteronuclear single quantum coherence (HSQC), ^1^H–^1^H correlation spectroscopy (^1^H–^1^H COSY), and heteronuclear multiple bond correlation spectroscopy (HMBC) experiments were performed on a Bruker Ascend 600 MHz NMR spectrometer equipped with 5 mm SmartProbeTM-600 MHz probe (Bruker, Billerica, MA, USA). The temperature of 300 K was used for the following experiments: ^1^H (pulse program “zg30”, TD = 65,536, NS = 128, SW = 19 ppm, O1P = 6 ppm, D1 = 1 s), ^13^C (pulse program “zgpg30”, TD = 65,536, NS = 15,000, DS = 16, SW = 220 ppm, O1P = 110 ppm, D1 = 2 s, P1 = 12 μs), 135 DEPT (pulse program “deptsp135”, TD = 65536, NS = 1024, DS = 8, SW = 220 ppm, O1P = 110 ppm, D1 = 2 s, P1 = 12 μs), TOCSY (pulse program “dipsi2gpphzs”, TD(F1) = 256, TD(F2) = 2048, NS = 16, DS = 16, D1 = 2 s, D9 = 90 ms), HSQC (pulse program “hsqcetgpsisp2.2”, TD(F1) = 256, TD(F2) = 2048, NS = 16, DS = 32, D1 = 1.5 s, CNST2 = 145 Hz), ^1^H–^1^H COSY (pulse program “cosygpmfqf”, TD(F1) = 128, TD(F2) = 2048, NS = 16, DS = 16, D1 = 2 s), HMBC (pulse program “hmbcgpndqf”, TD(F1) = 128, TD(F2) = 4096, NS = 16, DS = 16, D1 = 1.5 s, D16 = 0.2 ms). The peaks of ^1^H-NMR were calibrated at 4.70 ppm (the ^1^H signal of residual water in D2O), and all the peaks were picked and integrated using Topspin 4.1.4, Bruker, Billerica, MA, USA.

## 3. Results

### 3.1. Purification and Preparation of the SSPS

The carbohydrate could be easily distinguished from other substances by the PSPI (Figure 2). A broad band between 5 min and 10 min was detected simultaneously by RID and the UV-Vis detector. Therefore, the effluent at 5–10 min was collected to be freeze-dried and was considered to be SSPS-P. The content of the total carbohydrate in SSPS-P was determined to be 97.2% compared to 81.7% in SSPS using the phenol-sulfuric acid method. The rate of recovery of carbohydrate was detected as 86.5%.

There was a narrow band with a retention time of 12.6 min in RID, but it could not be detected by the UV-Vis detector. Since this band was not the target, it has not been analyzed in this work. Soybean polysaccharides are usually extracted and prepared using acidic water [20,22], which might combine with metal elements in the solution to form salts. The salt could not be completely removed during the alcohol precipitation process. These salts and other residual water-soluble small molecules in the extract cannot be separated by the SEC column which was used to separate polymers, thus showing a single band in RID.

### 3.2. Composition of the SSPS-P

According to HPLC analysis, there were two components in SSPS-P (Figure 3a). The main components exhibited a broad band between 13 min and 25 min, with the highest point corresponding to M_w_ of 354 KDa, accounting for approximately 80% of SSPS-P. In addition, there was a small band between 25 min and 30 min, and the M_w_ at the highest point was 16 KDa. SSPS-P was determined to be mainly composed of D-glc, D-gal, and L-ara, with a molar ratio of 0.02:2.08:1.01 (Figure 3b).

The FT-IR spectrum of SSPS-P exhibited typical polysaccharide structural characteristics (Figure 4). The strong broadband around 3325 cm^−1^ indicated many instances of O–H stretching in hydrogen bonds, which led to the strong inter- and intra-molecular interactions among polysaccharide chains [21]. The band around 2931 cm^−1^ was attributed to C–H stretching vibrational modes [23]. The band around 1612 cm^−1^ was assigned to the asymmetric stretching vibration of C=O, indicating the presence of the aldehyde groups or carboxyl groups, and several weak bands between 1418 cm^−1^ and 1350 cm^−1^ were attributed to the C–O stretching vibration and the O–H angular vibration [21]. The characteristic bands near 1075 cm^−1^ and 1045 cm^−1^ indicated the presence of D-galactopyranose (D-gal*p*) ring and L-arabinofuranose (L-ara*f*) ring, respectively [24,25]. In addition, it could be seen that the characteristic band at 891 cm^−1^ of SSPS-P suggested the presence of β-linked glycosyl residues [26,27].

NMR is the major approach to determine the chemical structure of polysaccharides. The ^1^H-NMR, ^13^C-NMR, 135 DEPT, TOCSY, HSQC, ^1^H-^1^H COSY, and HMBC data on SSPS-P were collected by NMR (Figure 5 and Figure 6). The ^1^H-NMR signals of SSPS-P were mainly concentrated between 3.4 and 5.4 ppm (Figure 5a). The chemical shift between 4.4 and 5.4 ppm was attributed to anomeric H-1 of glycosidic residues. There were five anomeric ^1^H signals at 4.58 ppm (A), 5.03 ppm (B), 5.10 ppm (C), 5.12 ppm (D), and 5.20 ppm (E) with the integrated value ratio of about 10:1:1:1:1 (Appendix A), which indicated five glycosidic residues in SSPS-P. Among them, the integrated value of the peak at 4.58 ppm was about twice the sum of the other four peaks, which indicated that it was dominant in SSPS-P. According to the result of the monosaccharides analysis, it might be attributed to a β-galactose residue [28]. The other four anomeric H signals at 5.03 ppm, 5.10 ppm, 5.12 ppm, and 5.20 ppm could be attributed to several substituted arabinose residues [28]. In ^1^H-NMR spectrum, the signal at 4.55 ppm was the anomeric hydrogen of β-glucose residue [29]. The integrated value of the signal at 4.55 ppm was approximately 1% of that at 4.58 ppm, which was consistent with the result of monosaccharide composition. Due to the low content of glucose residues in SSPS-P, it was difficult for us to accurately characterize.

As suggested by ^13^C-NMR signal analysis, the signals of SSPS-P were mainly concentrated within the range of 60–120 ppm (Figure 5b). Although there might be a C=O group in SSPS-P suggested by the characteristic band of 1612 cm^−1^ in FT-IR, there were no signals at 6–8 ppm in the ^1^H NMR or 170–190 ppm in the ^13^C-NMR spectrum [30], and it was consistent with the results of monosaccharide composition analysis. The weak signals at 102.83 ppm were the C1 of β-glc residue [29], which was difficult to characterize due to the low content. The major anomeric carbon signal peak was located at 104.36 ppm, which indicated the C1 of β-gal*p* residue. Other weak anomeric carbon signal peaks located at 106.40 ppm, 106.91 ppm, 107.09 ppm, and 107.45 ppm suggested the α-L-ara*f* residues [30]. According to the 135 DEPT (Figure 5c), the spectral analysis suggested that the peaks of 60.74 ppm and 60.86 ppm were negative peaks, which indicated methylene (-CH_2_) and could be attributed to the C6 of β-galactose residues and C5 of α-furanosyl residues (Figure 5c).

The anomeric carbon signal of 104.36 ppm with the corresponding anomeric hydrogen signal 4.58 ppm in residue A determined by HSQC spectrum (Figure 6b) was attributed to →4)-β-gal*p*-(1→ according to the report [23]. The cross peaks of H1–H2 (4.58/3.62 ppm), H2–H3 (3.62/3.71 ppm), H3–H4 (3.71/4.11 ppm), H4–H5 (4.11/3.66 ppm), and H5–H6 (3.66/3.78,3.67 ppm) can be determined using ^1^H–^1^H COSY spectrum (Figure 6c). Therefore, it was speculated that the H1, H2, H3, H4, H5, and H6 peaks of residue A could be assigned to the chemical shifts of 4.58 ppm, 3.62 ppm, 3.71 ppm, 4.11 ppm, 3.66 ppm, 3.78 ppm, and 3.67 ppm, respectively. Through the HSQC spectrum, it was suggested that the corresponding carbon peaks C1–C6 were 104.36 ppm, 71.82 ppm, 73.30 ppm, 77.66 ppm, 74.50 ppm, and 60.74 ppm, respectively.

The anomeric ^1^H-NMR signal corresponding to the peak of 107.45 ppm was 5.03 ppm in residue B according to HSQC (Figure 6b), which can be assigned to →3)-α-L-ara*f*-(1→ [31]. The H2-H5 can be determined to be 4.08 ppm, 3.90 ppm, 4.26 ppm, and 3.73/3.88 ppm by TOCSY (Figure 6a) and H-H COSY (Figure 6c). The signals assigned to C2-C5 also can be determined to be 81.26 ppm, 83.95 ppm, 84.91 ppm, and 61.11 ppm.

The anomeric ^1^H-NMR signal and ^13^C-NMR signal in residue C were at 107.09 ppm and 5.10 ppm, respectively, according to HSQC spectrum (Figure 6b). The signals of H2-H5 could be determined to be 4.08 ppm, 3.91 ppm, 3.99 ppm, and 3.78/3.67 ppm by TOCSY (Figure 6a) and ^1^H-^1^H COSY (Figure 6c), and the corresponding signals of C2-C5 were determined to be 81.10 ppm, 76.75 ppm, 83.83 ppm, and 60.86 ppm. The structure of residue C was assigned to T-α-L-ara*f*-(1→ [32].

Residue D could be assigned to→2)-α-L-ara*f*-(1→according to the signals of ^1^H-NMR (Figure 5a), ^13^C-NMR (Figure 5b), and HSQC spectrum (Figure 6b) and the literature [33]. The signals of C1/H1, C2/H2, C3/H3, and C5/H5 were at 106.91 ppm/5.12 ppm, 81.27 ppm/4.08 ppm and 76.72 ppm/3.91 ppm, and 60.86 ppm/3.78,3.67 ppm. Due to the low content of the residue, the signals of C4/H4 could not be observed.

The anomeric ^1^H-NMR signal and ^13^C-NMR signal in residue E was at 106.40 ppm and 5.20 ppm, respectively, according to HSQC spectrum (Figure 6b). The signals of 4.28 ppm and 4.20 ppm could be attributed to H2 and H3 according to the ^1^H-^1^H COSY spectrum (Figure 6c). However, the signals of H4-H5 and C2-C5 could not be observed due to the low content. According to the literature [34], residue E might be→3,5)-α-ara*f*-(1→.

1D NMR signals could provide information about glycoside residues, while 2D NMR can be used to assign the signals to ^13^C and ^1^H in these residues. The main ^13^C and ^1^H signals on SSPS-P were summarized in Table 1. The glycosidic bond signals and the connection relationship between glycosidic residues of SSPS-P could be determined by the HMBC spectrum (Figure 6d). The related signal peak between the anomeric carbon (A_C1_:104.36 ppm) glycosidic bond and A_H4_ (4.11 ppm) suggested the sugar chain composed of →4)-β-gal*p*-(1→ unit. The related peak between 107.09 ppm (C_C1_) and 4.08 ppm (D_H2_) suggested the T-α-L-ara*f*-(1→2)-α-L-ara*f*-(1→ unit.

Based on the above, SSPS-P was mainly a sugar chain composed of →4)-β-gal*p*-(1→unit. The units of T-α-L-ara*f*-(1→2)-α-L-ara*f*-(1→, →3)-α-L-ara*f*-(1→ and →3,5)-α-ara*f*-(1→ also were detected.

## 4. Discussion

PSPI could achieve the purpose of automated and rapid preparation of SSPS-P. Since our PSPI was developed on the basis of a semi-preparative high-performance liquid chromatograph, the amount of polysaccharides prepared could only be used for structural characterization and biological activity analysis. However, large-scale purification and preparation of polysaccharides could be achieved by adding split-flow valves and post-column derivative modules in the post-column flow path of a production line. Although the SSPS-P prepared by PSPI in this study achieved the purpose of increasing polysaccharide’s purity and yield, it contained multiple fragments due to the selected chromatography column. More efficient chromatographic packing could achieve the separation and collection of different polysaccharide fragments and could also be used for the separation and preparation of other polysaccharides from multiple sources.

SSPS-P with the M_w_ of 354 KDa had a structure with the backbone of →4)-β-gal*p*-(1→ unit and α-L-ara*f* residues, which was consistent with the reported soybean seed polysaccharides extracted using hot water [35]. There might be galacturonan with low content according to the FT-IR spectrum. The structure of α-1,4-galacturonic acid branched by β-1,4-galactan and α-1,3- or α-1,5-arabinan chains [20] has been reported. However, the monosaccharide composition and NMR spectrum did not support this conclusion. In fact, the hemiacetal structure of terminal sugar residues can be detected in FT-IR spectroscopy, and many polysaccharides [21,36] without glycuronate structures also have strong absorption bands around 1600 cm^−1^. Therefore, the peak at 1612 cm^−1^ in the FT-IR spectrum might be attributed to a very small amount of galacturonic acid and the hemiacetal of the terminal sugar groups.

## 5. Conclusions

In summary, based on the post-column split-flow and post-column derivatization schemes of PSPI, SSPS-P was distinguished from other substances and prepared. The content of the total carbohydrate in SSPS-P was achieved to be 97.2% compared to the 81.7% of SSPS crude sample, and the rate of recovery of carbohydrate was up to 86.5%. SSPS-P was a polysaccharide with the M_w_ of 354 KDa and had a structure with the backbone of →4)-β-gal*p*-(1→ unit. The α-L-ara*f* residues were also detected in the form of T-α-L-ara*f*-(1→2)-α-L-ara*f*-(1→, →3)-α-L-ara*f*-(1→ and →3,5)-α-ara*f*-(1→. It might be a polysaccharide with the →4)-β-gal*p*-(1→ as backbone and with the T-α-L-ara*f*-(1→2)-α-L-ara*f*-(1→ and →3)-α-L-ara*f*-(1→ and →3,5)-α-ara*f*-(1→ as side chains.

## Figures and Tables

**Figure 1 polymers-17-00480-f001:**
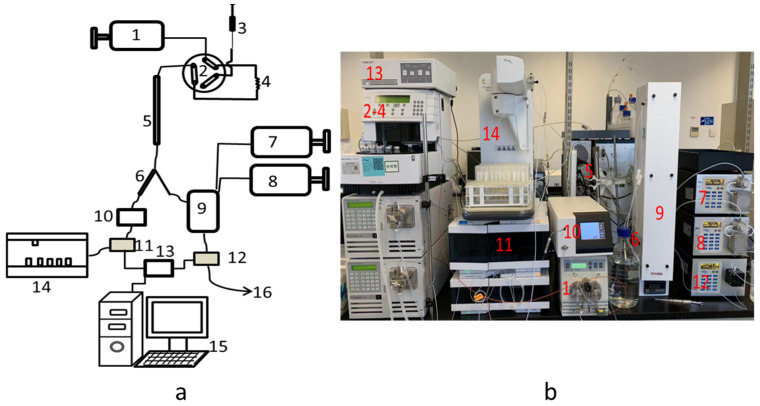
Schematic diagram of PSPI (**a**) and schematic presentation of the experimental design (**b**). In both (**a**) and (**b**), 1: high-pressure pump; 2: six-port valve; 3: sample injection needle; 4: quantitative loop; 5: chromatographic column; 6: splitter; 7: pump for concentrated sulfuric acid; 8: pump for 6% Phenol solution; 9: derivatization device; 10: pressure regulator; 11: RID; 12: UV-Vis detector; 13: digital-to-analog converter; 14: flow fraction collector; 15: the control system; 16: waste.

**Figure 2 polymers-17-00480-f002:**
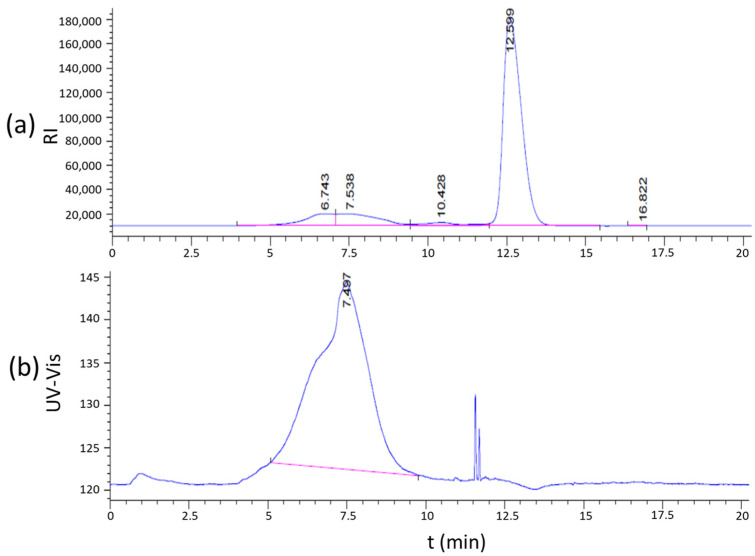
Chromatograms of SSPS in RID (**a**) and UV-Vis detector (**b**) detected using PSPI.

**Figure 3 polymers-17-00480-f003:**
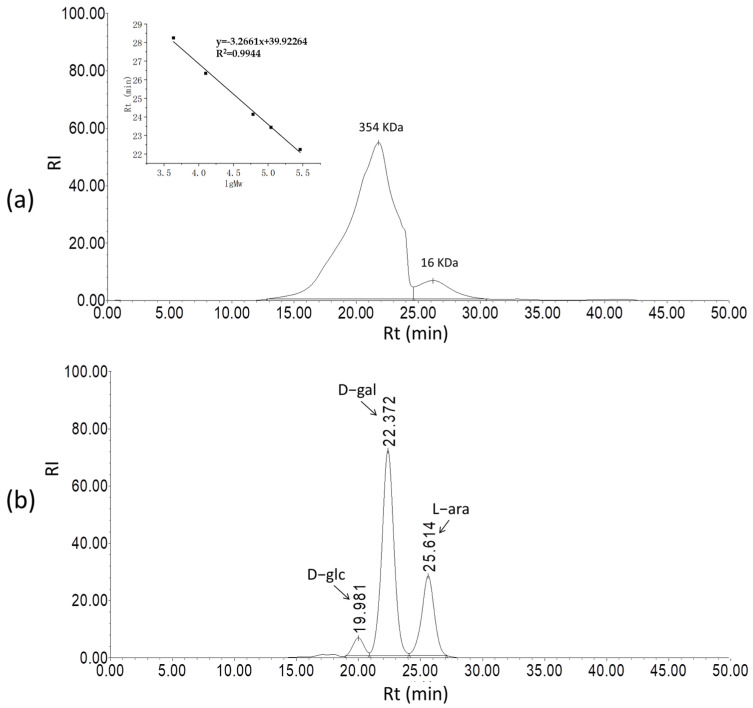
HPLC chromatograms of M_w_ (**a**) and monosaccharide composition (**b**) of SSPS-P. The graph inserted in (**a**) is the correlation curve between the logarithm of Mw and retention time, used to calculate the molecular weight of polysaccharides. The bands indicated by the arrow were determined to be D-glc, D-gal and L-ara, respectively at the retention time of 19.981 min, 22.372 min and 25.614 min.

**Figure 4 polymers-17-00480-f004:**
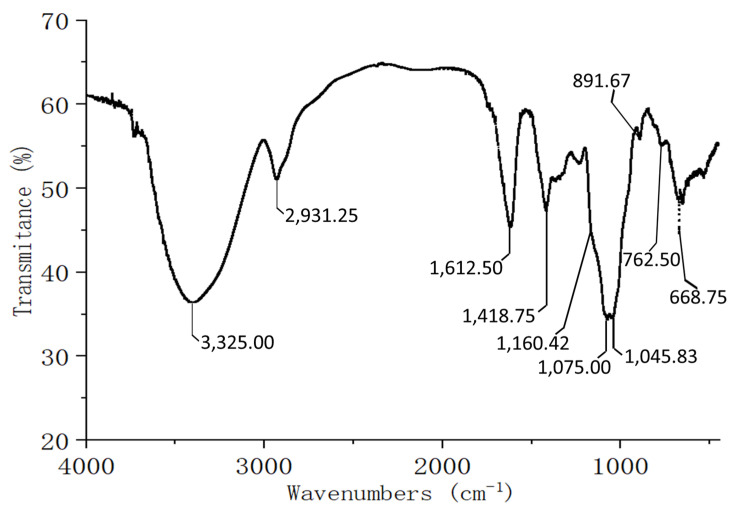
The FT-IR spectrum of SSPS-P.

**Figure 5 polymers-17-00480-f005:**
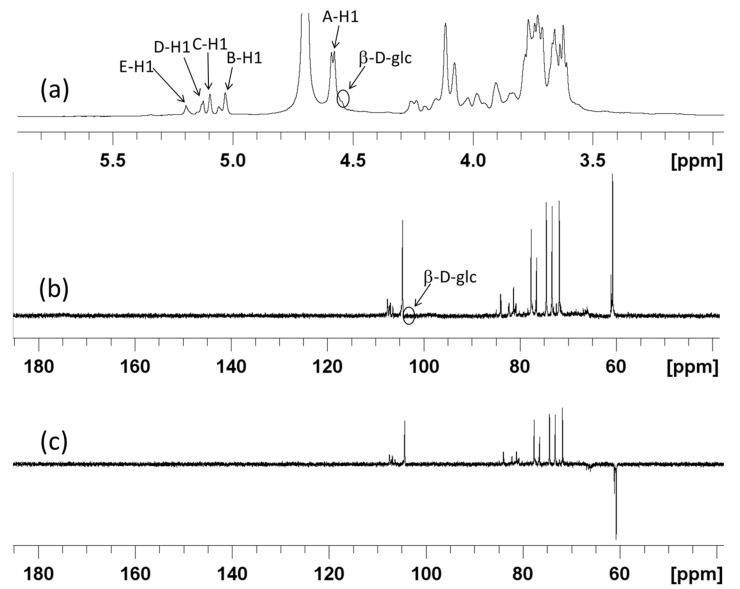
^1^H-NMR (**a**), ^13^C-NMR (**b**), and 135 DEPT (**c**) spectra of BSPS-P. The peaks attributed to the protons (or carbons) were indicated by the arrows.

**Figure 6 polymers-17-00480-f006:**
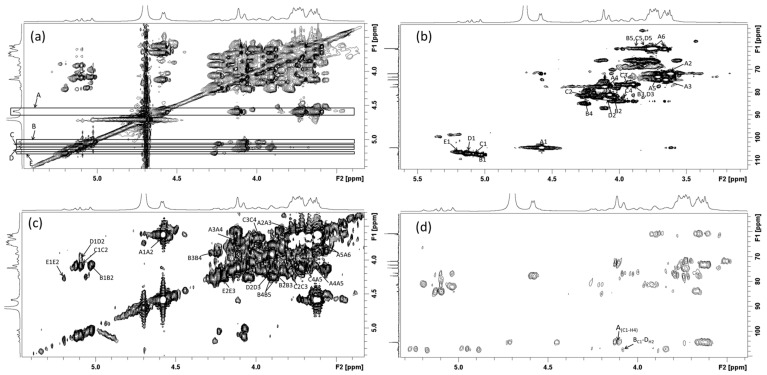
TOCSY (**a**), HSQC (**b**), and ^1^H-^1^H COSY (**c**) and HMBC (**d**) spectra of BSPS-P. The cross peaks in the figure were attributed to the atoms indicated by the arrows, such as the signal of C1-H1 cross peak in residue C was indicated by an arrow in (**b**) and labeled as C1.

**Table 1 polymers-17-00480-t001:** ^1^H and ^13^C NMR chemical shift assignments (in ppm) of SSPS-P.

Residue		C1	C2	C3	C4	C5	C6
		H1	H2	H3	H4	H5	H6
A	→4)-β-gal*p*-(1→	104.36	71.82	73.30	77.66	74.50	60.74
		4.58	3.62	3.71	4.11	3.66	3.78/3.67
B	→ 3)-α-L-ara*f*-(1→	107.45	81.26	83.95	84.91	61.11	/
		5.03	4.08	3.90	4.26	3.73/3.88	/
C	T-α-L-ara*f*-(1→	107.09	81.10	76.75	83.83	60.86	/
		5.10	4.08	3.91	3.99	3.78/3.67	/
D	→2)-α-L-ara*f*-(1→	106.91	81.27	76.72	na *	60.86	/
		5.12	4.08	3.91	na	3.78,3.67	/
E	Might be →3,5)-α-ara*f*-(1→	106.40	na^1^	na	na	na	/
		5.20	4.28	4.20	na	na	/

*: could not be determined.

## Data Availability

Data are contained within the article and Appendix A.

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
