# Peer review of "Application of a Polysaccharide Purification Instrument—The Preparation and Characterization of Soybean Soluble Polysaccharide"

_polymers, 2025, doi:10.3390/polym17040480_

Round 1
Reviewer 1 Report
Comments and Suggestions for Authors
Section 2.1 - is the quality of the sulphuric acid important? Was the phenol dissolved in water?
Section 2.2 - how do you correct for the different times spent in each of the pathways?
Section 3.1 - do you know what the peak at 12.6 mins is? Do you have a signal from excess derivatising agent?
Section 3.2 - is the composition data the average of both peaks from the SEC or just one of them? Try to avoid commas in your numbers as this is ambiguous. Try to be consistent is it D-gal or D-Gal? Can you suggest why you don't see the C=O peak in 13CNMR? What does "reversal peaks" mean? Do you know what the molecule looks like? Are the arabinose residues side chains, for example.
Reference # 16 - is incorrect
Comments on the Quality of English Language
There are a few issues in the English language which could be improved, for example, in the last sentence of the induction it should be "purified"
Author Response
Comments 1: Section 2.1 - is the quality of the sulphuric acid important? Was the phenol dissolved in water?
Reply: Due to the sulphuric acid in the DFPath was much more than the amount required for derivatization reaction, the remaining sulphuric acid would flow through the UV-Vis detection. The quality of the sulphuric acid might influent the signals of the UV-Vis detection, thus the superior purity grade was used. 6% phenol solution was used as one of the derivatization reagents in the DFPath. The 6% phenol solution was prepared by dissolving every 6 grams of phenol in 100 mL of water. At Section 2.1, the preparation of the phenol solution was added.
Comments 2: Section 2.2 - how do you correct for the different times spent in each of the pathways?
Reply: After the PSPI was developed, Dextran10000 (Shanghai Macklin Biochemical Tech-nology Co., Ltd) was used for the test of the equipment. The retention time of Dex-tran10000 was used to calibrate the peak time of DFPath and CFPath. The lengths of the pipelines in the two flow paths were optimized to ensure that the retention time of Dextran10000 in both flow paths remains consistent. The second and the third paragraphs were added in Section 2.2.
Comments 3: Section 3.1 - do you know what the peak at 12.6 mins is? Do you have a signal from excess derivatising agent?
Reply: The chromatographic peak at 12.6 minutes was not analyzed in our work. Since these substances cannot be detected by UV-Vis detector in DFPath, it was not a carbohydrate. We speculated that it was salt and other water soluble small molecules. Soy polysaccharides are usually extracted and prepared using acidic water, which combines with metal elements in the solution to form salts. The salt could not be completely removed during the alcohol precipitation process. These salts and other residual water-soluble small molecules in the extract cannot be separated by SEC column which was used to separate polymers, thus showing a single peak in RID.
Before injection of the sample, the derivatization reagents (sulphuric acid and phenol solution) were excess of in the DFPath, and the signal on the UV-Vis detector was a flat straight line without chromatographic peaks. The instrument has been passed a third-party verification in 2023 (Verification Number: CBJ-WP-BG-001-2023) by Shanghai WEIPU Testing Technology Group Co.,Ltd
Comments 4: Section 3.2 - is the composition data the average of both peaks from the SEC or just one of them? Try to avoid commas in your numbers as this is ambiguous. Try to be consistent is it D-gal or D-Gal? Can you suggest why you don't see the C=O peak in 13CNMR? What does "reversal peaks" mean? Do you know what the molecule looks like? Are the arabinose residues side chains, for example.
Reply:
(1) The retention time of each chromatographic peak was based on the vertex of that peak in the SEC. Therefore, this description was modified to be “The main components exhibited a broad peak between 13 min and 25 min, with the highest point corresponding to Mw of 354 KDa, accounting for approximately 80% of SSPS-P. In addition, there was a small peak between 25 min and 30 min, and the Mw at the highest point was 16 KDa” in the revised manuscript.
(2) The commas in the numbers were removed and all the numbers in the text were checked.
(3) The spelling of all sugar residues has been modified to consistency.
(4) Due to the low abundance of 13C in nature (approximately 1.1%) and the lack of NOE from surrounding hydrogen atoms, the signal value of C=O in 13C-NMR is very weak. Due to the content of acetyl ester in polysaccharides is low; it is difficult to be detected. In addition, the hemiacetal structure of terminal sugar residues also can be detected in FT-IR spectroscopy, and many polysaccharides(such as https://doi.org/10.1016/j.ijbiomac.2022.07.043 and http://dx.doi.org/10.1016/j.ijbiomac.2015.10.075) without glycuronate structures also have strong absorption bands around 1600cm-1. Therefore, the peak at 1612cm-1 in the FT-IR spectrum might be attributed to a very small amount of galacturonic acid and the hemiacetal of the terminal sugar groups.
(5) In the section 3.2, the "reversal peaks" is the negative peak in 135 DEPT experiment, which indicates the methylene (-CH2) in the molecule. The "reversal peaks" was corrected to be “negative peak” in the revised manuscript.
(6) According to the reference, the molecule of the SSPS might be a polysaccharide with the →4)-β-galp-(1→ as backbone, and with the T-a-L-araf-(1→2)-a-L-araf-(1→ and →3)-α-L-araf-(1→ and →3,5)-α-araf-(1→ as side chains.
Comments 5: Reference # 16 - is incorrect
Reply: The reference #16 was corrected and all the other references were checked.
Comments 6: There are a few issues in the English language which could be improved, for example, in the last sentence of the induction it should be "purified"
Reply: The entire text was revised and improved, many errors were also corrected in the revised namuscript.
Reviewer 2 Report
Comments and Suggestions for Authors
The paper, entitled Application of a Polysaccharide Purification Instrument - The Preparation and Characterization of Soybean Soluble Polysaccharides, is primarily concerned with the application of a polysaccharide purification instrument (PSPI) developed using the post-column split-flow and post-column derivatization schemes described in their earlier work. Here, PSPI was used for the separation and purification of soybean soluble polysaccharides (SSPS), and purified SSPS (SSPS-P) was obtained. The characterisation of SSPS-P was also carried out using spectroscopic techniques. The only problem I see in this work concerns some segments of the characterisation. Firstly, HPLC suggests that the composition of SSPS-P should contain D-Glc, but this was not confirmed by other spectroscopic techniques. Also, the absorption band at 1612 cm-1 visible in the IR spectrum, which the authors attribute to the asymmetric stretching vibration of C=O, suggesting the presence of acetyl ester, was not confirmed by 1 and 2D NMR spectra. The authors should explain the above-mentioned inconsistencies in these results of spectroscopic techniques in the characterisation of SSPS-P a little more clearly.
Author Response
Comments: The paper, entitled Application of a Polysaccharide Purification Instrument - The Preparation and Characterization of Soybean Soluble Polysaccharides, is primarily concerned with the application of a polysaccharide purification instrument (PSPI) developed using the post-column split-flow and post-column derivatization schemes described in their earlier work. Here, PSPI was used for the separation and purification of soybean soluble polysaccharides (SSPS), and purified SSPS (SSPS-P) was obtained. The characterisation of SSPS-P was also carried out using spectroscopic techniques. The only problem I see in this work concerns some segments of the characterisation. Firstly, HPLC suggests that the composition of SSPS-P should contain D-Glc, but this was not confirmed by other spectroscopic techniques. Also, the absorption band at 1612 cm-1 visible in the IR spectrum, which the authors attribute to the asymmetric stretching vibration of C=O, suggesting the presence of acetyl ester, was not confirmed by 1 and 2D NMR spectra. The authors should explain the above-mentioned inconsistencies in these results of spectroscopic techniques in the characterisation of SSPS-P a little more clearly.
Reply: There were weak signals in 1H-NMR and 13C-NMR. Due to the very low content of D-glc, it is difficult to fully characterize. The hemiacetal structure of terminal sugar residues also can be detected in FT-IR spectroscopy, and many polysaccharides(such as https://doi.org/10.1016/j.ijbiomac.2022.07.043 and http://dx.doi.org/10.1016/j.ijbiomac.2015.10.075) without glycuronate structures also have strong absorption bands around 1600cm-1. Therefore, the peak at 1612cm-1 in the FT-IR spectrum might be attributed to a very small amount of galacturonic acid and the hemiacetal of the terminal sugar groups.